# Intimate Partner Violence in *Khaliji* Women: A Review of the Frequency and Related Factors

**DOI:** 10.3390/ijerph20136241

**Published:** 2023-06-28

**Authors:** Maisa H. Al Kiyumi, Asma Said Al Shidhani, Hana Al Sumri, Yaqoub Al Saidi, Amira Al Harrasi, Maryam Al Kiyumi, Sanaa Al Sumri, Aseel Al Toubi, Maithili Shetty, Samir Al-Adawi

**Affiliations:** 1Department of Family Medicine and Public Health, Sultan Qaboos University Hospital, Sultan Qaboos University, Muscat 123, Oman; asmash@squ.edu.om (A.S.A.S.); alsumry@squ.edu.om (H.A.S.); yaqoubs@squ.edu.om (Y.A.S.); amirarh@squ.edu.om (A.A.H.); m.alkiyumi1@squ.edu.om (M.A.K.); drshs@squ.edu.om (S.A.S.); 2Department of Behavioral Medicine, College of Medicine & Health Sciences, Sultan Qaboos University, Muscat 123, Oman; aseel.altoubi@gmail.com (A.A.T.); maithili.shetty97@gmail.com (M.S.)

**Keywords:** intimate partner violence, domestic violence, prevalence rates, forms of IPV (physical, sexual, emotional, economic), risk factors, cultural attitudes towards violence and gender roles, GCC, Arab, *Khaliji*

## Abstract

The Gulf Cooperation Council (GCC), locally known as *Khaliji*, is a group of six Arab nations, including Saudi Arabia, Bahrain, Kuwait, Oman, Qatar, and the United Arab Emirates (UAE). Intimate partner violence (IPV) is a significant public health concern in the aforementioned region, but research that synthesises this trend has remained scarce. The present narrative review examines existing research on the prevalence and frequency of IPV among *Khaliji* women who inhabit the GCC nations. This review synthesised studies on physical and sexual violence, emotional abuse, and controlling behaviours perpetrated by an intimate partner. The prevalence rates of IPV among *Khaliji* women were observed to be high: women reported facing different types of abuse from their partners, namely physical (7–71%), sexual (3.7–81%), financial (21.3–26%), and psychological (7.5–89%), which is a culmination of controlling behaviour (36.8%), emotional violence (22–69%), and social violence (34%). Existing studies in the GCC region suggest that the most endorsed IPV was psychological abuse (89%), followed by sexual violence (81%). Qualitative analysis of the content of associated factors resulted in four significant descriptors, such as victim demographics, sociocultural factors, socioeconomic factors, and perpetrator-related issues. Research on IPV is still in its nascent stages, with very few studies emanating from the GCC region. The way forward will require developing culturally appropriate interventions that address the unique risk factors for IPV among the *Khaliji* population, strengthening institutional responses, and increasing awareness and social support for victims of IPV.

## 1. Introduction

Intimate partner violence (IPV) is a form of domestic violence that occurs between intimate partners, such as spouses, boyfriends, or girlfriends. The World Health Organisation (WHO) defines intimate partner violence as “any behaviour within an intimate relationship that causes physical, psychological, or sexual harm to those in the relationship”, with controlling behaviour appearing as a form of abuse [1,2]. In some sociocultural settings around the world, women are not granted the same legal rights and protections as men. This may lead to the de facto treatment of women as property or objects that are controlled by male family members or husbands [3]. These predicaments are sometimes reinforced by sociocultural views that promote a collective mindset in which individual autonomy is not socially sanctioned [4]. Although such collective and interdependent teachings can foster IPV, the ‘geography’ of IPV is not limited to just collectivistic societies, individualistic societies, or even patriarchal or matriarchal societies, for that matter [5]. It is possible that cultural and social contexts and gender roles and expectations, coupled with legal frameworks and social support systems, could shape the frequency and factors associated with IPV. To date, these factors have received scant attention.

IPV can take many forms, including physical, sexual, emotional, or psychological abuse. The consequences of IPV have been reported in the literature [6]. Sequalae include injuries such as bruises, broken bones, head injuries, chronic pain including headaches and back pain, sexually transmitted infections, and unintended pregnancies. The broader physical sequelae could include long-term health problems such as cardiovascular disease, chronic pain, and gastrointestinal disorders. The corpus of literature has also highlighted the emotional impact of IPV on survivors, including all spectrums of psychiatric disorders. These often trigger various problems of externalisation and internalisation behaviours, including low self-esteem, feelings of worthlessness, self-blame, difficulty trusting others and forming new relationships, guilt, shame, humiliation, and suicidal thoughts or behaviours [7]. There is also evidence to suggest that IPV can also impact and trigger subtle but debilitating cognitive impairment manifesting as difficulty in concentrating and making decisions, difficulties in learning and remembering, increased distraction, and impulsivity [8]. The effects of IPV have also been reported to cascade across the social and economic sphere, including isolation from one’s social network, family and friends; loss of social support and resources; financial dependence and financial difficulties; and homelessness or insecure housing [7]. IPV can have negative impacts on children and is often associated with behavioural problems, emotional difficulties, and poor academic performance among offspring [9]. Given the diverse negative impacts of IPV, it would require equally diversified support and interventions to address the same.

A systematic review of IPV in the Middle East/North Africa (MENA) region by Elghossain, Bott, and Akik [10] reported the results of multiple population-based studies from seven countries. The frequency of IPV ranged from 6% to as high as 59% for physical violence, 3% to 40% for sexual violence, and 5% to 91% for emotional/psychological violence. More recently, Kisa, Gungor, and Kisa [11] published their scoping review of the trends in the MENA region. The authors reported that physical injuries are a common sequel of IPV. Factors associated with IPV included age, education, length of marriage, previous experiences of childhood abuse or witnessing family violence, geographical area (rural location), and family income. Interestingly, both the systematic review and the scoping review did not proceed with a meta-analysis. It is not clear whether such an omission stems from the heterogeneity of the design, methods, or outcome measures of existing studies. The conspicuously absent data from Arabian Gulf countries (except Saudi Arabia) is another frequently observed trend [10]. It is essential that a critical review of the trends in the GCC is performed even if the studies appear to employ suboptimal methodology. The rationale for this is that, although less robust, these studies can still provide valuable information on IPV experiences in contexts that have not been extensively studied before. This can help identify unique risk and protective factors and cultural or social factors that can influence IPV in these regions. Although less robust methodologies may not provide the same level of precision or accuracy as data from more rigorous research designs, they can still offer valuable information that can inform future research and policy. For example, such studies can identify important areas for further investigation or highlight the need for culturally appropriate interventions to address IPV in different contexts.

The term ‘*Khaliji*’ or ‘Khaligi’ refers to the Gulf region of the Arabian Peninsula. This includes countries that make up the six nations that make up the economic and political alliance that is the Gulf Cooperation Council (GCC) [12]: Oman, Saudi Arabia, Kuwait, Bahrain, Qatar, and the United Arab Emirates (UAE). In the present discourse, *Khaliji* women are women from the GCC or who identify with its culture and traditions [4]. One key aspect of the population structure of the GCC is its urbanisation. The region is rapidly urbanising, with an increasing number of people moving to cities in search of better economic opportunities. According to the World Bank, the majority of the population now lives in urban areas [13]. *Khaliji* populations have diverse family structures shaped by various cultural, religious, and historical factors [14]. Studies have suggested that *Khaliji* populations have family structures that fall into extended or tribal families and are often polygamous and consanguineous. The latter is firmly rooted in sociocultural practices, but there is a dissenting view [15]. With increased acculturation and their general move from rural to urban areas, nuclear families are becoming increasingly common in tightly woven patriarchal societies; Every individual is expected and supposed to have a family, ethnicity, or tribal connection [14]. As is often the case with patriarchal societies, men are typically seen as breadwinners and decision makers, while women are expected to prioritise their roles as wives and mothers [4]. However, recent reports suggest that the GCC has undergone a social transformation that would directly affect the well-being of women. A recent World Bank Annual Report entitled Women, Business and the Law 2020 (WBL) aims to assess laws and regulations that affect women’s economic opportunity in 190 economies [16]. Saudi Arabia has implemented groundbreaking reforms to such an extent that it has now been lauded as a ‘global top reformer’. A similar observation has been made in other GCC countries as well. Although women’s empowerment is an important goal, it is crucial to address the topology and associated factors of IPV to further ensure that women have the necessary resources, support, and protection to navigate the challenges of the two worlds: traditional and modern. A review synthesising the emerging trends in IPV directed toward *Khaliji* women will help set the background for greater scrutiny of the topic and more evidence-based prevention and intervention. The present discourse aims to fill this gap in the existing literature.

The objectives of this review were twofold. The first was to examine the frequency of IPV among *Khaliji* women. A second related objective was to quantify the factors associated with IPV. These associated factors have the potential to identify possible avenues for the prevention and mitigation of IPV. A secondary objective of this review was to examine the frequency of IPV and its associated factors through the lens of sociocultural practises, as these influences play an important role in shaping various aspects of human behaviour.

## 2. Materials and Methods

The current review covered articles up to April 20, 2023, as it served as a narrative assessment of the epidemiology and correlates of IPV in the GCC. Inclusion criteria allowed the consideration of studies that (i) were conducted in the GCC region, including countries such as Saudi Arabia, Bahrain, Kuwait, Oman, Qatar, and the UAE; (ii) had IPV as its primary focus; (iii) examined the prevalence and frequency of physical and sexual violence, emotional abuse, and controlling behaviours perpetrated by an intimate partner; (iv) were quantitative or qualitative; (v) provided data on the prevalence rates or qualitative insights into IPV experiences; and (vi) were published in English. Articles were excluded if the articles (i) focused on research populations outside of the GCC region, (ii) did not specifically focus on IPV among *Khaliji* women, (iii) did not provide data or insight on the prevalence or frequency of IPV or its associated factors, or (iv) were published in languages other than English.

The articles were searched using keywords that reflect the IPV spectrum. Keywords to specify IPV were “intimate partner violence”, “domestic violence”, “dating violence”, “spousal abuse”, “gender-based violence”, “battering”, “relationship violence”, and ‘family violence’. Academic databases such as PsychINFO, Scopus, Google Scholar (to accommodate grey literature), and PubMed/Medline were used to retrieve articles. According to some sociocultural practises around the world, intimate relationships tend to be associated with those relationships that are legally bound by marriage. The sexual mores and forms of intimate relationships are becoming increasingly diverse [17]. Some related terms of IPV include “intimate partner aggression”, “interpersonal violence”, “partner violence”, “intimate terrorism”, and ‘coercive control’ [18]. In the present discourse, these related concepts will be used interchangeably when a reference to IPV is made.

Keywords used for the English articles were divided into four levels as follows; each level had to be included in the search: epidemiology [level 1] OR “point prevalence” OR ‘period prevalence’ OR “lifetime prevalence” OR ‘incidence’ OR “prevalence” OR “survey” OR ‘incidence rates’ OR “incidence rates” OR “epidemiological studies” OR ‘longitudinal studies’ OR “longitudinal studies” OR ‘population-based surveys’ OR “population-based surveys” OR “community-based surveys” OR “victim surveys” OR “perpetrator surveys” AND [Level 2] Oman OR ‘Saudi Arabia’ OR “Qatar” OR “UAE” OR “Kuwait” OR “Bahrain” AND [Level 3] “determinants”: the related term for associated factors, risk factors, predictors, and correlates. This term is often used in epidemiology and public health research to refer to any factor or variable that contributes to the occurrence or distribution of a health outcome, such as a disease or condition. Determinants can be classified into various categories, including demographic (eg age, gender), socioeconomic (e.g., income, education), behavioural (e.g., smoking, physical activity) and environmental factors (e.g., access to health care) [19].

The search methodology did not impose any restrictions on the time period of the articles considered. Relevant published papers were also examined, but only those relating specifically to the current region of interest.

The extraction is shown in the flow chart below (Figure 1). Complete versions of the articles were downloaded once the evaluators verified that their titles and abstracts met the inclusion criteria. After further exclusion of articles that did not meet the inclusion criteria, five independent authors (M.K., A.S., H.S., Y.S. and A.H.) manually searched the reference lists of all articles for any articles that may have been missed during the initial search process.

The accumulated articles were then analysed to identify any variables, indicators, determinants or factors associated with IPV in Bahrain, Kuwait, UAE, Qatar, Saudi Arabia, and Oman. The associated factors accrued from the identified articles were then tabulated and narrated into meaningful descriptors using a qualitative content analysis approach [20]. Qualitative content analysis is a research method used to analyse textual data by systematically categorising, coding, and interpreting its meaning. In this approach, the associated factors are identified and then categorised into different themes or codes based on their similarities and differences. These themes or codes are then narrated in meaningful descriptors that provide insight into the underlying patterns or relationships in the data. This analysis has specifically employed an iterative method, which means that the authors continually review and refine the coding framework as more associated factors are identified. Once the coding is complete, the researcher reviews the coded data and identifies the patterns, themes, or categories that emerge from the data. Similar codes are grouped to form meaningful categories. As shown in the table, the associated factors in the accrued data constituted themes pertinent to ‘demographic factors’, ‘sociocultural factors’, ‘socioeconomic factors’, and “perpetrator-related factors”.

## 3. Results

The literature search yielded 14 articles that met the inclusion criteria for the prescribed period up to 20 April 2023, without a lower time limit that covered the frequency and associated factors of IPV in the six countries of the GCC. In the present search, the time frame was from 2009 to 2022. Saudi Arabia appeared to have the highest number of IPV articles (*n* = 7), followed by Bahrain (*n* = 2) and the UAE (*n* = 2), Oman (*n* = 1) Qatar (*n* = 1), and Kuwait (*n* = 1).

### 3.1. Frequency

The IPV frequency is shown in Table 1, and for each country, the rate is highlighted below in tandem. In Saudi Arabia, studies examining IPV violence of intimate partners found that the prevalence of women exposed to lifetime violence ranged from 11.9% to 44.8%. Furthermore, among the frequency of different types of violence revealed, 9–45.5% of women faced physical abuse and 6.9–19.2% faced sexual abuse. The highest percentage of domestic violence among partners was reported to be psychological abuse, which was a culmination of mental and emotional abuse, standing at 35.9% and 22–69%, respectively.

Regarding the incidence rate of IPV studies in Oman, almost 28.8% of women had experienced lifetime abuse by their partners, 21% were exposed to emotional abuse and 18% experienced physical abuse. Furthermore, approximately 10.1% of the women also reported being emotionally and physically abused by their respective partners.

Among women in the UAE, the prevalence of psychological abuse ranged from 7.5% to 46%. Furthermore, the majority of women were also reported to be physically and sexually abused, with a prevalence of 7–32% and 3.7–14%, respectively.

A study in Qatar explored the incidence of reported violence among a sample population of 2787 women and found that nearly 12.4% of married women had experienced lifetime abuse from their partners.

Studies that investigated IPV among women attending primary care centres in Bahrain showed that almost 30.1% to 71.7% of women reported having ‘ever experienced’ domestic violence. The type of violence varied between emotional abuse (60%), physical abuse (32.9%), sexual abuse (13.8%), and financial abuse (21.3%).

A study among Kuwaiti women showed that the highest reported type of violence was sexual abuse, with a prevalence of 81%, followed by psychological abuse (75%) and physical abuse (71%).

### 3.2. Associated Factors

The second objective of this study was to examine the determinants of IPV among *Khaliji* women. Identified articles were analysed using qualitative content analysis, where associated factors were grouped as themes and narrated as meaningful descriptors. Qualitative content analysis resulted in broader and more comprehensive descriptors, namely, (i) demographic, (ii) social/cultural, (iii) socioeconomic, and (iv) perpetrator-related factors (Table 2).

The first, broadly associated factors, are those related to the victim and the perpetrator’s demographic background. Significant factors emerging among studies in the GCC included the age of the impacted person, the length of time they were married to their partner, victims being overweight or obese, being a *Khaliji* woman who grew up with parents who were divorced, being a *Khaliji* woman who has been divorced and remarried, situations in which the perpetrator lived with a widowed mother, and the education of *Khaliji* women.

Regarding sociocultural-associated factors, existing studies in the GCC suggest that women who have lived in polygamous marriages or households that sanction violence in a marital context and those who report a lack of support from society are likely to experience IPV.

Socioeconomic factors contributing to IPV included lack of tangible support, insufficient income, perceived lack of financial independence, and financial control of the husband.

The final set of associated factors was those relating to the characteristics of the perpetrator. Husbands with adverse childhood experiences, low education, unemployment, or certain occupations (such as being in the military) may be at increased risk of perpetrating domestic violence. Similarly, husbands engaged in substance misuse, with poor self-regulation, with poor mental health outcomes, and with autocratic tendencies may also be at increased risk of engaging in abusive behaviour.

## 4. Discussion

An economy now primarily based on oil and gas exports, the once desolate Arabian Peninsula has undergone a rapid transformation within the past few decades. Although efforts are being made to diversify their economies away from oil and gas, many of the GCC countries have been internationally lauded for undergoing this rapid economic transformation. According to Smith [35], the money earned from the oil industry has propelled the development at a remarkable pace, achieving in less than two decades what took Europe a thousand years. Although there are some subcultural differences, the six GCC countries nevertheless share several similarities, including history, language, religion, culture, economy, and politics, which, in turn, have fostered a sense of common identity and cooperation [36].

Rapid modernisation of the GCC in recent years has resulted in significant material progress in its population and a decline in regressive practises that harm women, such as female genital mutilation, child marriage, and honour killing [37,38]. However, much like most of the world, the challenges of being part of a primarily patriarchal society are common in the lives of *Khaliji* women. Patriarchal societies often expect women to conform to traditional gender roles and expectations, and their primary role is conventionally limited to being a wife, mother, and caregiver of the household. Although such traditional gender roles are not inherent in patriarchal societies alone and can exist in various social and cultural contexts, in the context of IPV, they are likely to create power imbalances and massive gender inequality. This, in turn, perpetuates harmful beliefs and attitudes toward women that could lead to an acceptance or normalisation of IPV. In some of such societies, women are often considered property of their husbands or male relatives and may be subjected to various forms of violence, discrimination, and oppression. Despite such misgivings, most of the atavistic views toward women certainly have the potential to be changed by laws and emerging ideas of empowerment. Many countries in the MENA region have agreed to comply with the necessary sanctions to reduce the incidence of IPV against women as part of their effort to adhere to the stipulations of Sustainable Development Goal (SDG) 5.2.1 and other related indicators [39]. However, it is important to note that the existence of a law or policy does not necessarily guarantee effective implementation or translation into meaningful change in reality [40].

The prevalence of IPV in the world appears to be quite significant. Approximately 13% of women who have ever been in a committed relationship and are between the ages of 15 and 49 have suffered physical or sexual IPV in the past year [41]. The lifetime prevalence of IPV varies by region, ranging from 20% in the Western Pacific to 49% in Oceania and Central Sub-Saharan Africa [42]. A study involving 124,838 women found that the prevalence rates of any type of IPV during pregnancy ranged from 1.8% to 99.5%, depending on the study [43]. In general, around 30% (nearly 1 in 3) or 736 million women around the world have suffered IPV or non-partner sexual violence [44]. The magnitude reported from the broader Eastern Mediterranean or MENA region has been featured in an international survey. Approximately 67% of IPV victims were in the age range of 15 to 34 years [43].

One of the most remarkable characteristics of the MENA region is the youth population. According to the United Nations, the average age in the region is about 25 years, significantly younger than the global average of 31 years. This is largely due to the high fertility rate and the large proportion of the population under the age of 30 years [45]. Young people are often associated with spearheading social change, including changes in values, aspirations, and social norms. Younger generations in this regard, may alter the dynamics of personal relationships and the resulting prevalence of IPV when they question or redefine conventional social norms [46,47,48,49]. To address and prevent violence in relationships, it is important to understand how shifting social dynamics and IPV interact.

In the GCC, “the family is the basic building block of society”, and “the family institution remains strong” (p. 352) [15]. In certain countries of the GCC, it is mandatory to undergo premarital screening programmes to ensure the compatibility of couples and to identify any genetic disorders that could potentially affect their children [50]. This implies that the region is making a concerted effort to increase the quality of life of *Khaliji* women and their offspring. According to the Ministry of Health, a “healthy marriage guarantees to prevent family members from hereditary and infectious diseases; thus building a happy and stable family” [51]. Therefore, quantifying the magnitude and associated factors of IPV, which the late Secretary of State of the UN, Kofi Annan, called “… the most shameful violation of human rights” [52], can contribute to fulfilling the goal of a stable family life and better quality of life for *Khaliji* women, in particular.

According to a 2013 WHO report [53], at the global level, more than one in three women have experienced physical or sexual assault from a partner or sexual violence from a non-partner, implying that IPV is a phenomenon that transcends culture, ethnicity, and geography. In the Americas, a systematic review of IPV frequency among studies from 24 nations suggested that 14% to 17% of women in Brazil, Panama and Uruguay had suffered IPV or its variants on the spectrum [54]. Data from Bolivia suggested the highest IPV while a general trend of lower IPV rates emerged in North America compared to South America. In Europe, the IPV rate fluctuated in a complex way with the type of IPV considered. In general, Barbier, Chariot, and Lefèvre reported that 51.7% of women had suffered from IPV [55]. The study concluded that the rates were likely to be underreported. In Africa, a systematic review by Roman and Frantz reported that IPV ranged from approximately 26.5% to 48% [46]. Regarding Asia and Pacific region, Jewkes et al. [56] reported a lifetime prevalence of 27.5% to 67.4% in Bangladesh, Cambodia, China, Indonesia, Sri Lanka, and Papua New Guinea. In general, the magnitude of IPV in the GCC region appears to fall within the global ranges. The figures currently available from different continents could only be the tip of the iceberg due to underreporting, as IPV is a highly stigmatised and sensitive topic. Furthermore, the lack of standardised measurement tools, as different studies may use different definitions and measures of IPV, creates challenges in comparing and contrasting results between various studies. Sampling bias could be yet another confounder, as studies often rely on convenience samples, i.e., women seeking services from shelters or health clinics, which may not accurately represent the total population of women experiencing IPV. Another factor that hampers accurate measurement of the magnitude is cultural and linguistic barriers. In some cultures, the discussion of IPV may be too taboo, and language barriers may prevent survivors from seeking help or accurately describing their experiences. These barriers can significantly hinder the data collection process for the magnitude of IPV in that particular culture. For example, women responding to IPV assessment questionnaires from societies where ‘beating’ the wife is a norm rather than an exception may not perceive beating as a culturally devalued act. This lack of awareness can significantly affect how we capture the true extent of IPV within that culture [57].

The present review included 14 articles that met the inclusion criteria. These articles focused on the frequency and associated factors of IPV in the current region of interest, the six countries of the GCC. The articles were published between 2009 and 2022. Among these countries, Saudi Arabia had the highest number of articles (*n* = 7) discussing IPV in women. Bahrain and the UAE had two articles each, while Oman, Qatar and Kuwait each had one article discussing IPV. The present review suggests that *Khaliji* women have been documented to have physical problems, ranging from 7% to 71%. Other subtle but debilitating forms of IPV also appear to exist, including financial deprivation, which ranged from 21.3% to 26% of the sample surveyed, and psychological trauma, which ranged from 7.5% to 75% of the sample. Other IPV subtypes were also observed among *Khaliji* women, including controlling behaviour (36.8%), emotional abuse (22–69%), and social violence (34%). Almost higher than the global average is sexual violence (81%). As different studies have used different concepts and the resultant instruments to tap into IPV, the generalisation of the presently observed frequency of IPV is likely to be tampered with by such constraints. For example, of the 14 articles, 5 studies measured the frequency of IPV using their own or adapted instruments [22,25,29,30,31]. Two articles developed an instrument derived from the Norvold Abuse Questionnaire [28,32] and one from the World Health Organisation [23]. Six studies used previously established instruments that are known to solicit the presence of IPV, including the World Health Organisation Multi-country Instrument on Violence Against Women [27], the Intimate Partner Violence Against Women Questionnaire [24], the Norvold Domestic Abuse Questionnaire [21], the Severity of Violence against Women Scales [34], the short version of the Women Abuse Screening Tool [33], and the Women’s Experiences with the Battering Scale [34]. As discussed above, the reviewed studies employed heterogeneous assessment tools. This means that the present generalisation could echo the conundrum of ‘comparing apples with oranges’. Therefore, culturally sensitive instruments, equipped with items that keep the option of international comparison open, will be essential.

### 4.1. Associated Factors

The second objective of the present review was to explore the factors associated with IPV among the 14 articles that met the currently defined inclusion criteria. Using qualitative content analysis [51], the current venture resulted in four comprehensive descriptors that included topics relevant to factors relating to demographic, social/cultural, socioeconomic, and perpetrator-related factors. These associated factors were recapitulated in conjunction with the existing literature.

The age of women and the duration of marriage have been documented and were found to be significantly associated factors in the studies that emerged from the GCC. Studies in Saudi Arabia, Bahrain, and the UAE showed that the duration of marriage was strongly associated with IPV. Women who had been married for a duration ranging from 10 to 20 years were more prone to experiencing IPV compared to those who had been married for less than 10 years or more than 20 years. When looking at data from Saudi Arabia and Bahrain, risk factors for IPV increased after a decade of marriage and lasted for 20 years. Furthermore, it was also reported to be correlated with IPV. This implies that women aged 30 to 40 years reported experiencing more IPV.

Previous studies have suggested that, on the one hand, women who have experienced violence from their intimate partners are at increased risk of developing obesity [58], and on the other hand, obesity in some populations around the world tends to be stigmatised, which in turn could contribute to negative body image and self-esteem [59]. Recent lifestyle changes that contribute to maintaining a sedentary lifestyle have led to an increased risk of obesity [60]. In the Arab population, overweight has previously been reported to be considered a symbol of status and a desirable trait in women. Historically, studies have indicated that, for example, in some GCC societies, there is a cultural preference for women to be voluptuous. This trait was often even enhanced by force-feeding (or gavage), but such practices appear to have eroded with recent modernisation [61]. However, this perception has been under scrutiny in recent years, with an increasing recognition that obesity has negative health effects and the “globalisation of the ideal lean body’ driven by popular culture [62]. Therefore, the emerging ideal body image in the GCC has been used among abusive husbands to shame their partners. The surveys conducted by the National Epidemiological Health Survey [63] showed that the prevalence of obesity among *Khaliji* women is significantly higher than among men. In Kuwait, for example, the obesity rate for women stood at 47.9%, while a 34.6% prevalence was reported among men. Similarly, in Qatar and Saudi Arabia, approximately 45.3% and 44% of women were classified as obese, respectively. This is almost twice the rate of men in these countries. As obesity is on the rise among *Khaliji* women and husbands are more likely to use their wife’s body size to ostracise them, studies are needed to unravel this emerging trend.

The present review suggests that divorce appears to be a significantly associated factor in the context of IPV, especially among women with divorced parents or who had remarried. *Khaliji* women who grew up with divorced parents may be more vulnerable to IPV due to a variety of factors, including unresolved trauma, difficulty trusting others, and lack of positive role models for healthy relationships, as documented in other populations [64]. These views echo psychodynamic theories suggesting the role of early childhood experiences in shaping adult relationships and behaviours. Consequently, childhood experiences can lead to unresolved conflicts and emotional wounds that can manifest in adult relationships and contribute to dysfunctional behaviours [65].

Divorce is often stigmatised in society, especially for women, who can be seen as failures or shamed for not being able to maintain their marriages [66]. Many families would rather see their daughters stay married even if their husbands are abusive than deal with the shame that a divorce would entail. Divorced Arab women tend to be isolated from society, blamed and made to feel sinful [59]. Future studies should examine whether divorce represents a risk factor for IPV or if it is the result of multiple factors.

Another demographic factor associated with IPV among *Khaliji* women is living with a widowed mother-in-law. Being ‘family-centric’ [5], many traditional families in GCC communities tend to live with their extended family [67,68]. In some extended families, the household can have many generations under one roof. Conflicts between in-laws, such as the mother- and daughter-in-law, become inevitable and can lead to misunderstandings and conflicts due to a generational gap and differences in beliefs. In some cases, the mother-in-law may feel threatened by the role of the daughter-in-law in the family and her influence on her son, leading to feelings of jealousy and resentment [69]. In such an enmeshed family setting, tensions could build up, with the husband sometimes choosing to side with the mother. This, in turn, could further escalate the tensions and ultimately result in the husband’s practise of IPV against his wife.

Research conducted among European populations has indicated that women’s education can act as a safeguard against IPV. This is supported by evidence suggesting that as women’s educational attainment increases, their vulnerability to experiencing IPV decreases [70]. More recently, women have certainly seen some of the benefits of the spread of education in GCC countries, often empowering them to escape abusive husbands. As the present study suggests, it appears that a lack of education, as previously reported in other populations, makes some vulnerable *Khaliji* women to experience an increased risk of IPV. In most of the GCC countries, girls’ education is compulsory and the enrolment rates of girls in primary and secondary education have increased significantly [71]. Similar trends have been observed in the higher education sector where, according to the public media, women make up the majority of university students in Qatar [72]. Oman has also seen a positive educational trend with a recently observed ‘feminisation’ of health services [73]. It remains to be seen whether women’s empowerment has had a direct positive impact on reducing IPV. Therefore, more studies are warranted.

Sociocultural factors play an important role in shaping attitudes toward IPV. A sociocultural factor that is prevalent in the GCC is patriarchal communities where traditional beliefs promote men as the head of the household and relegate women to the peripheries, expecting them to remain subservient. This further promotes the idea that men have the right to control and discipline their wives, which can lead to justifications for violence in the context of ‘maintaining discipline’ or “correcting” behaviour [74]. Another factor is the prevalence of gender roles, where the roles of women are primarily at home. Women who do not adhere to these gender roles can be seen as disobedient and may be subjected to violence as a form of punishment. The question then arises of whether polygamy is another sociocultural factor that has the potential to increase the risk of IPV. Although studies exploring this relationship are lacking in the population of the GCC [75], in a study of 16 sub-Saharan African countries, Ahinkorah [76] reported that the percentage of women involved in polygamous marriages ranged from 1.6% to 40%. The author reported that polygamy was associated with a higher likelihood of experiencing IPV. Intuitively, one can predict that the presence of multiple wives could create a sense of rivalry and competition, which may lead to conflict and the violence and resultant IPV. The sociocultural factors that contribute to IPV are likely to be complex and deeply ingrained. Addressing these factors requires a comprehensive approach that includes efforts to change attitudes, promote gender equality, and provide education and support to IPV victims.

Various factors have emerged to be related to the perpetrator of IPV. Studies have shown that there is a strong association between adverse childhood experiences (ACEs) and IPV in adulthood [77]. ACEs can involve a variety of types of childhood abuse, neglect, and family dysfunction. Factors such as sadness, anxiety, impulsivity, and problematic drinking can act as mediators in the association between ACE and IPV [78]. Furthermore, substance use (e.g., ‘self-medication’) can often be practiced to cope with the trauma of childhood abuse [79]. An associated factor that echoes the trend elsewhere is the increased risk of IPV among women who have intimate relationships with soldiers. Military services that involve exposure to combat, deployment-related stressors, and mental health problems have been speculated to contribute to this increased risk of IPV [80].

The aforementioned discussion of factors associated with IPV prominently features women who have experienced IPV. These factors include descriptors related to demographics, society/culture, socioeconomics, and the perpetrators themselves. Present risk factors appear to echo international trends and broadly echo those identified by the Centres for Disease Control and Prevention (CDC) [81].

### 4.2. Limitations

One limitation of this review is that the included quality of the articles was not assessed. Therefore, more research on IPV in the present region of interest. Ideally, critical evaluations, such as a systematic review and meta-analysis, will be essential to examine the present trend. Second, it is unclear whether the magnitude of IPV reported among *Khaliji* women is representative of the entire GCC region, given the large expatriate population and the fact that there is a maldistribution of the studies, with only Saudi Arabia having a majority of the studies. Some of the studies in the GCC were conducted primarily in specific facilities (e.g., healthcare settings), leading to a possible sampling bias. In the future, a more robust community-level survey, including expatriate communities, is needed. Third, cultural and religious norms that only allow relationships between married couples may also mask the true prevalence of IPV. As alluded to earlier, the types of intimate relationships and sexual mores are changing. Therefore, the magnitude of IPV emerging in intimate relationships maintained outside the traditional realm of marriage may not be brought to the forefront. As reported elsewhere, interdependent societies, such as those in the GCC, tend to aspire to maintain family harmony and privacy, as well as to avoid shame and stigma. This may prevent women from reporting IPV and seeking help [82]. Despite this view, the present review has suggested high-frequency psychological abuse (89%), followed by sexual violence (81%). However, a concerted effort will be needed to quantify the frequency and risk factors for IPV using culture-sensitive measures to avoid data spuriousness.

Despite the limitations mentioned above, there is a need to (i) create a culturally sensitive classification system for IPV, (ii) conduct more rigorous research to accurately determine the extent of IPV, (iii) improve education and prevention efforts to reduce the occurrence of IPV, and (iv) expand and improve services for those affected by IPV. These points are further elaborated on below.

The first step is to create a culturally sensitive classification system for IPV and to take into account the unique circumstances and factors that can contribute to IPV. To date, there have been various emphases on what constitutes IPV, including physical, sexual, psychological, or emotional, and economic violence. It is not clear whether such a taxonomy could lead to misunderstanding or an oversimplification of the complex issue of IPV in a different culture. Baker and Dwairy [82] have argued that cultural taboos surrounding sexuality and sex can make it difficult for victims to come forward and seek help in a collectivist society. Individuals in such societies (as those often seen in the GCC) are patterned to operate in an interdependent and collective framework. Thus, much cherished individual autonomy, commonly observed as part of the legal system in Euro-American societies, is not largely reinforced in traditional communities in the Global South [83]. Related to this, it has been reported that when stress occurs in the *Khaliji* population, the affected individual is likely to experience distress in its somatic sense rather than the “psychic pain” often observed and documented in the Euro-American population [84]. If this view has heuristic value, this would imply that the perception and sequelae of IPV are likely to be experienced within this sociocultural context. Qualitative research can help uncover these cultural factors and provide information on how they influence the occurrence and perception of IPV within such a specific cultural context. Therefore, the use of multiple sources of evidence to build a more comprehensive understanding of IPV is essential, if not paramount.

Second, to obtain a more accurate picture of the magnitude and scope of IPV, more robust research studies that use reliable and valid measures and diverse samples are needed. The population of the GCC is known to be ’social media savvy’. Elsewhere, McCauley et al. [85] have suggested that one way to gauge the “pulse” of society is to look at what is covered on social media platforms [85]. Consequently, what is reported on social media platforms has the ability to serve as an influential means to initiate a public discourse on the actualities of IPV and can provide a platform for IPV victims to share their experiences and receive support. In a study that highlighted the importance of online social support for IPV victims, Homan et al. [86] suggested that machine learning and other computational methods can be used to analyse social media trends. The high number of social media users in the GCC, along with their diverse demographics, could serve as a valuable tool to better measure the incidence of IPV. It may also be helpful to identify potential ways to mitigate IPV.

Third, in conjunction with research efforts, it is crucial to increase education and prevention efforts to reduce the occurrence of IPV. This includes raising awareness about the signs and effects of IPV, as well as promoting healthy relationship skills and communication. Reducing IPV in the GGC requires multifaceted approaches. The religious inclination of many communities here could mean that a way of appealing to those communities might be simply reminding them that, according to Roesch et al. [87], Prophet Mohammed has advised Muslims to treat women with kindness and be the best to their wives. Motivating believers to remember that the Quran also commands that husbands be gentle with their wives, even if there are problems in their relationship, and that it prohibits mistreatment and forcing women to do anything against their will, could be helpful: “I command you to be kind to women” … “The best of you is the best to his family (wife)” (Sunan al-Tirmidhi); “O you who believe! You are forbidden to inherit women against their will. Also, do not treat them with harshness” (Quran 4:19).

In addition to socioculturally relevant teaching, there is a need to increase public education on what constitutes abuse and how to identify warning signs [70,88]. To date, there is a dearth of studies that directly address the attitude toward and awareness of IPV among Arab populations, and the GCC is no exception. Raising awareness of IPV can help reduce the stigma associated with it and encourage victims to seek help. Furthermore, to consolidate this, it would be essential, if not paramount, to strengthen laws and policies according to which survivors of IPV can be aided and perpetrators of abuse are made accountable for their actions. This includes criminalising all forms of IPV, providing resources to victims, and ensuring that perpetrators are prosecuted. Developing policies and laws that protect victims of IPV and hold perpetrators accountable can help reduce the incidence.

Lastly, it is important to expand and improve services for those affected by IPV. This includes providing accessible and culturally sensitive support services. The GCC must expand and further develop services to help IPV survivors, including law enforcement, healthcare providers, and social services. These organisations should receive training on how to work with diverse populations and provide culturally sensitive services that meet the needs of people of different ethnic and cultural backgrounds, given the diversity within the GCC region [89]. In many societies where support and resources are widely available, it appears that for victims of abusive relationships, a range of services must be established to rebuild their lives, including culturally sensitive counselling and emergency shelters. The GCC countries have been lauded internationally for promoting women’s empowerment, but this sudden shift in practises and beliefs could itself present unforeseen challenges for women who are now having to deal with the duality of their own two worlds: the traditional and the modern. Therefore, reducing IPV in the GGC requires a multifaceted approach that involves raising awareness, strengthening laws and policies, providing support services, engaging men and boys, and promoting a culturally sensitive path of empowerment that has the most potential to reduce IPV.

## 5. Conclusions

*Khaliji* women have recently witnessed that their part of the world has undergone rapid economic and social changes. The GCC has been internationally lauded for having secured unprecedented material progress in the past few decades, and women have demonstrated the empowerment precipitated by the spread of education and the judicial system that safeguards their well-being. However, like their counterparts in the Global South, in the Global North, *Khaliji* women still endure intense forms of gender-based violence. In the present analysis, the frequency of physical abuse ranged from 7% to 71%, sexual abuse from 3.7% to 81%, financial abuse from 21.3% to 26%, and psychological abuse from 7.5% to 89%. By any standard, such figures are severe, but the generalisation of this study should be reviewed with caution, given that there is a maldistribution of research and the field itself appears to be in its nascent stages. Various associated factors, including demographics and sociocultural, socioeconomic and perpetrator-related factors, were found to be significant and would require further examination. As a final analysis, it was concluded that IPV is a complex issue that requires a multifaceted approach that includes legal, social, and cultural interventions. Efforts to address IPV in the GCC must focus on raising awareness, providing support services to survivors, strengthening legal frameworks, and challenging gender norms that contribute to perpetuating violence. More studies are warranted in the future in order to establish a more evidence-based and informed approach.

## Figures and Tables

**Figure 1 ijerph-20-06241-f001:**
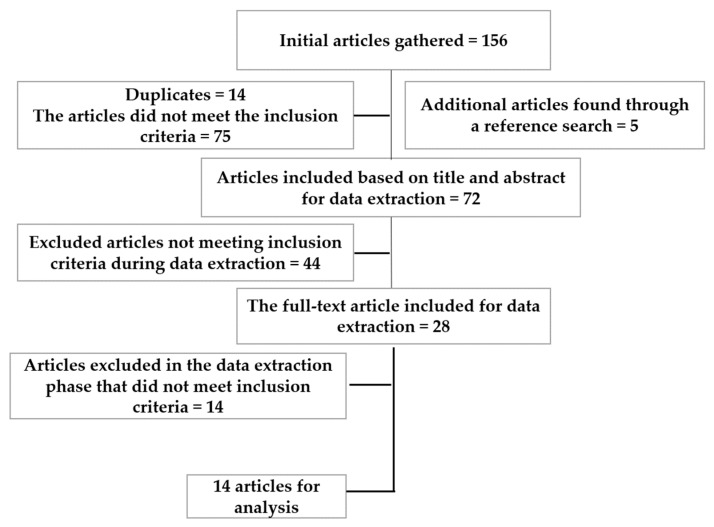
Article extraction flowchart: inclusion and exclusion process.

**Table 1 ijerph-20-06241-t001:** Frequency of and factors associated with intimate partner violence in *Khaliji* women.

Country	Authors	IPV Detection Tool	*n*	Catchment Area	Prevalence
Saudi Arabia	Wali et al. [21]	Norvold Domestic Abuse Questionnaire	*n* = 1845	Primary Health Care, Western Region	Lifetime prevalence = 33.24%
					Psychological abuse = 48.47%
					Physical abuse = 34.77%
					Sexual abuse = 16.75%
					All three types of abuse = 4.1%
	Afif et al. [22]	The authors’ self-developed questionnaire with input from WHO Multi-country Study	*n* = 2000	Al-Ahsa Oasis in the Eastern Province	Lifetime—39.3%
					Mental—35.9%
					Physical—17.9%
					Sexual—6.9%
	Alzahrani, Abaalkhail, and Ramadan [23]	Self-developed questionnaire with input from the World Health Organisation	*n* = 497	Primary Health Care, Taif City	Overall—11.9%
	Alquaiz et al. [24]	Modified from the Intimate Partner Violence Against Women Questionnaire developed by the World Health Organisation	*n* = 1883	Primary Health Care, Riyadh	Lifetime violence—43%
					Controlling behaviour—36.8%
					Emotional—22%
					Physical—9%
					Sexual—12.7%
	Barnawi [25]	Author’s self-developed questionnaire	*n* = 720	Al-Wazarat Primary Health Care, Riyadh	Overall—20%
					Emotional—69%
					Social—34%
					Financial—26%
					Physical—20%
					Sexual—10%
	Eldoseri and Sharps [26]	World Health Organisation Violence against Women Questionnaire (v.10.0)	*n* = 200	Primary Health Care, Jeddah City	Physical violence = 45.5%
	Abolfotouh and Almuneef [27]	WHO multi-country instrument on violence against women	*n* = 400	Primary Health Care, Riyadh	Overall—44.8%
					Physical—18.5%
					Emotional—25.5
					Sexual—19.2%
					Financial—25.3%
Oman	Al Kendi et al. [28]	The authors developed a self-developed questionnaire with input from the Norvold Domestic Abuse Questionnaire	*n* = 978	Primary Health Care, Muscat Governorate	Lifetime—28.8%
					Emotional—21%
					Physical—18%
					Emotional and physical—10.1%
UAE	Serkal et al. [29]	Self-developed questionnaire	*n* = 700	Primary Health Care, Dubai	Physical—7.14%
					Sexual—3.7%
					Psychological—7.5%
	AlMulla and Alothman [30]	Self-developed questionnaire	*n* = 920 married women	National community survey	Physical—32%
					Psychological—46%
					Sexual—14%
Qatar	Al-Ghanim [31]	Author’s self-developed questionnaire	*n* = 2787	A national tertiary education centre in Doha	Overall—2.22%
					Overall—12.4%
	Bubshait et al. [32]	Authors’ self-developed questionnaire with input from the Norvold Abuse Questionnaire	*n* = 602	Primary Health Care	Overall—30.1%
					Emotional—60%
					Physical—32.9%
					Sexual—13.8%
					Financial—21.3
	Al Ubaidi et al. [33]	Women Abuse Screening Tool—Short Version	*n* = 810 women	Primary Health Care	Overall—71.7%
Kuwait	Alsaleh [34]	The Severity of Violence Against Women Scales Women’s Experiences with Battering Scale	*n* = 1335 women, randomly selected community survey	Kuwait City	Physical—71%
					Sexual—81%
					Psychological—89%

**Table 2 ijerph-20-06241-t002:** A Qualitative Analysis of Factors Associated with Intimate Partner Violence Among *Khaliji* Women.

Four Significant Descriptors	Content of Associated Factors
Demographic Factors	Age of the victim and duration of the marriage
	Victims with above-average weight
	The victim has divorced parents
	The victim has gone through a divorce and remarried
	Perpetrator living with a widowed mother
	Education level of the victim
Sociocultural Factors	Victims living in a polygamous marriage
	Victims who live in households that sanction violence in a marital context
	Victims allege a lack of support from society
Socioeconomic Factors	The victim does not receive tangible support in her marriage
	The victim lives in a household with insufficient income
	Victim’s perceived lack of financial independence from the perpetrator
	The perpetrator controls the financial matters of the victim
Perpetrator-Related Factors	The perpetrator has a history of ACE (adverse childhood experience)
	The low educational level of the perpetrator
	Unemployed perpetrator
	Perpetrator’s employment in a military occupation
	Perpetrator’s smoking, alcohol use, and gambling habits
	A perpetrator with poor self-regulation, poor mental health outcome, and autocratic tendencies

## Data Availability

The authors confirm that the data supporting the findings of this study are available in the article.

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
