# Peer review of "Intimate Partner Violence in Khaliji Women: A Review of the Frequency and Related Factors"

_ijerph, 2023, doi:10.3390/ijerph20136241_

Round 1

Reviewer 1 Report

This paper is very interesting and it appears to add to the IPV literature on a specific population of Khaliji women. I do have a few comments/questions for the authors. 

Abstract

1. Line 25. There are a plethora of studies on IPV, so maybe change this sentence to reflect that you’re referring to studies on this specific area. 

Introduction

2. There’s discussion of the cultural differences, but it really isn’t until I got to the discussion that I understood the actual nuances of why this population of women is important to look into. More discussion of these factors in the front end would be helpful in setting up the significance of this study. 

3. Why do you expect this population to have distinct differences in IPV experiences? 

Results

4. Were there only 14 IPV articles from GCC countries? Or were some excluded? If some were excluded, please state why and how the sample size got to 14. If there were only 14 articles, make that clearer. 

Limitations

5. Line 430–why wasn’t this assessed?

Author Response

Dear Reviewer 1

We highly appreciate your constructive comments. Attached to this message, you will find a document containing a point-counterpoint format. This format has been prepared to address each of your inquiries.

Reviewer 2 Report

Dear Authors:

Thank you for the opportunity to review this paper entitled “ Intimate partner violence in the Khaliji women: a review of frequency and associated factors.”  The research topic has an essential value in the application and prevention of intimate partner violence, and good to research it. However, several significant problems need to be addressed.

First, in the method section.

1.   What were the inclusion criteria and exclusion criteria of this study?  

2.   A research flow chart should show the number of articles and the context of selection and deletion conditions in the research process.

3.   What’s the process of data collection? Who collected the data for this study? Have they been well trained? Does this research obtain approval by IRB?

4.   How to conduct qualitative content analysis? How many people participated in the evaluation of the articles? Are the people reviewing the article qualified? What is the basis for the assessment? If there are different evaluation results, how to make a decision?

5.   In Table 1, the scales of several articles were developed by the authors themselves, but they need to explain the reliability and validity of the scales. How to judge whether the measured results are credible 

Second, in the results section.

1.   Table 1 needs to be completed.

Last, it is more critical that intimate partner violence includes married partners and unmarried boyfriends and girlfriends, and the background and related factors of intimate partner violence for married and unmarried people are different. Therefore, it is inappropriate that the authors put them together and collectively refer to the incidence and associated factors of violence in intimate relationships9

Author Response

Dear Reviewer 2

We highly appreciate your constructive comments. Attached to this message, you will find a document containing a point-counterpoint format. This format has been prepared to address each of your inquiries.

Reviewer 3 Report

Thank you for the opportunity to review the manuscript. Overall, a current topic for a broader readership and further exploration of this topic is certainly unique, especially to examine the frequency of IPV among Khaliji women and to quantify the factors associated with IPV in the Gulf Cooperation Council.

A few questions / comments and suggestions:

In your discussion, in Line 240-247, what are their traditional gender roles and expectations, relevant to the study is not clear.

In Line 263-264, how to set the youthful population and how to have the most notable features, relevant to the study is not clear.

In Line 302-304, more elaboration of the culture’s barriers, relevant to the study is not clear.

In Line 305-306, clearly present which 14 articles, relevant to the study is not clear.

In Line 315-317, clearly present which different studies, relevant to the study is not clear.

In Line 403-408, clearly present which different related studies, relevant to the study is not clear.

Thank you for the opportunity to review the manuscript. Overall, a current topic for a broader readership and further exploration of this topic is certainly unique, especially to examine the frequency of IPV among Khaliji women and to quantify the factors associated with IPV in the Gulf Cooperation Council.

A few questions / comments and suggestions:

In your discussion, in Line 240-247, what are their traditional gender roles and expectations, relevant to the study is not clear.

In Line 263-264, how to set the youthful population and how to have the most notable features, relevant to the study is not clear.

In Line 302-304, more elaboration of the culture’s barriers, relevant to the study is not clear.

In Line 305-306, clearly present which 14 articles, relevant to the study is not clear.

In Line 315-317, clearly present which different studies, relevant to the study is not clear.

In Line 403-408, clearly present which different related studies, relevant to the study is not clear.

Author Response

Dear Reviewer 3

We highly appreciate your constructive comments. Attached to this message, you will find a document containing a point-counterpoint format. This format has been prepared to address each of your inquiries.

Round 2

Reviewer 1 Report

Thank you for addressing the comments. This paper makes a clear contribution to the literature. 

Author Response

Dear Colleague

Thank you for your positive feedback and kind words regarding the paper.

 If you have any further comments or suggestions, please do not hesitate to share them with us.

Best regards,

Samir & the rest of the authors